# Reward type influences adults' rejections of inequality in a task designed for children

**Katherine McAuliffe** [ID] [1]*, **Natalie Benjamin**[2], **Felix Warneken**[3]

**1** Department of Psychology and Neuroscience, Boston College, Chestnut Hill, MA, United States of America, **2** Department of Psychology, Marquette University, Milwaukee, WI, United States of America, **3** Department of Psychology, University of Michigan, Ann Arbor, MI, United States of America

* katherine.mcauliffe.2@bc.edu

**Data Availability Statement:** Data have been uploaded as supporting information.

**Funding:** The author(s) received no specific funding for this work.

## Abstract

In the context of economic games, adults sacrifice money to avoid unequal outcomes, showing so-called inequity aversion. Child-friendly adaptations of these games have shown that children, too, show inequity aversion. Moreover, inequity aversion shows a clear developmental trajectory, with young children rejecting only disadvantageously unequal distributions and older children rejecting both disadvantageously and advantageously unequal distributions. However, based on existing work, it is difficult to compare adult and child responses to inequity because (1) adapting economic games to make them child-friendly may importantly alter the dynamics of the fairness interaction and (2) adult work typically uses abstract rewards such as money while work with children typically uses more concrete rewards like candy, stickers or toys. Here we adapted the Inequity Game—a paradigm designed to study children's responses to inequality in isolation from other concerns—to test inequity aversion in adults (*N* = 104 pairs). We manipulated whether participants made decisions about concrete rewards (candy) or abstract rewards (tokens that could be traded in for money). We found that, like children, adults rejected unequal payoffs in this task. Additionally, we found that reward type mattered: adults rejected disadvantageous—but not advantageous—monetary distributions, yet rejected both disadvantageous and advantageous candy distributions. These findings allow us to draw clearer comparisons across child and adult responses to unfairness and help paint a fuller picture of inequity aversion in humans.

## Introduction

Fairness concerns are a lynchpin of cooperative societies: people across cultures conform to fairness norms [1] and enforce those norms in others [2–4]. Underscoring the importance of these concerns for humans, fairness emerges early in children and shows marked development across childhood. Expectations of fairness emerge within the first two years of life [5–7]. By the preschool years, children are beginning to share resources fairly with others in some contexts, such as after joint collaboration [8], and become generally more likely to conform to fairness norms by late childhood [9, 10]. This general developmental pattern is apparent across diverse cultural groups, with some variation in the age of onset and the degree to which fairness norms such as merit or equality prevail [11].

**Competing interests:** The authors have declared that no competing interests exist.

One context in which fairness concerns are clearly revealed is in people's responses to unequal resource distributions. Adults show an aversion to resource inequalities—so-called inequity aversion—and will sacrifice personal gain to avoid inequality [12–14]. Inequity aversion in adults is seen, to varying degrees, in both directions of inequity: people have a strong aversion to receiving less than others, disadvantageous inequity aversion, and a weaker aversion to receiving more than others, advantageous inequity aversion [14]. For instance, in a seminal paper by Dawes and colleagues [12], adult participants paid to reduce the payoffs of players who had randomly received more than others (showing disadvantageous inequity aversion) and paid to augment the payoffs of players who had randomly received less than others (showing advantageous inequity aversion). Additionally, adults show a willingness to pay for fairness in a range of other economic contexts, including the Dictator and Ultimatum Games (reviewed below), pointing to the importance of fairness concerns in driving decision-making in the context of resource distribution.

Work with adults has highlighted the importance of fairness for humans, yet adult work can tell us little about the origins of our fairness preferences. To understand the full picture of where these preferences come from and what factors help shape their expression, it is essential to combine data from adults and children. However, a problem inherent in studying fairness across both children and adults is that it is often difficult to compare child and adult data directly due to the different experimental methods used. In the present paper, we address this issue by conducting a test of disadvantageous as well as advantageous inequity aversion, originally designed for children, with adult participants. From a methodological perspective, we highlight the strengths and limitations of using the same task to study fairness across adults and children. In our introduction, we first briefly review economic game approaches for studying fairness in children. Next, we describe work that has crossed different participant-group boundaries, for instance conducting the same study with both adults and children or with both a nonhuman animal species and humans. Finally, we describe the approach taken in the present study and highlight our major aims and predictions.

## Studying fairness in children using economic games

While distributive fairness in children has been studied using a wide range of methodologies [10, 15–23], one especially fruitful approach has been to create child-friendly versions of economic games that were originally designed to measure fairness preferences in adults. These games are useful in large part because they tend to involve real stakes, allowing participants to make decisions that have material consequences for themselves and others. For instance, the Dictator Game has been used successfully with children across a wide age range [16, 19, 24–26]. In this game, one participant can unilaterally share resources with another person, providing a measure of generosity and fairness. The Ultimatum Game, in which Player A can offer some proportion of an endowment to Player B who can then accept or reject the offer, affecting the payoffs of both parties, has been used as a measure of second-party punishment of unfairness [27–29] and, when compared with donations in the Dictator Game, a measure of strategic reasoning in children [27]. As these two examples illustrate, participants in these games must sacrifice personal gain in order to be fair. Compared with other widely-used developmental methods, which often involve asking children to endorse or enact different distributions as a third party in hypothetical scenarios (e.g., [21, 30]), economic games in which real resources are at stake may thus come closer to revealing what children *actually* do as opposed to what they think they *ought* to do, a distinction which has been labelled the knowledge-behavior gap [31].

Another benefit of using economic games to study fairness in children is that, in principle, we should be able to make direct comparisons between work with children and adults using the same game, giving us a fuller picture of the developmental trajectory of fairness. Indeed, these games have been used successfully to compare decision-making among older children, adolescents, and adults. For instance, Gummerum and Chu [32] conducted a mini-Ultimatum Game, a game in which a proposer chooses between one of two pre-set distributions and a responder can either accept or reject (i.e., punish) their choice, with 8-, 12-, and 15-year-olds and adults. They found that, with age, participants became more sensitive to the choice constraints imposed on the proposer, rejecting fewer unfair offers when the proposer had no choice but to be unfair. Sutter [23] conducted a mini-Ultimatum Game with 7- to 10-year-olds, teenagers and adults and found a similar pattern of results: younger participants were relatively more focused on outcomes and less on the proposers' intentions than were older participants. These studies have helped shed light on the development of responses to unfairness across childhood, adolescence and adulthood. However, they are constrained by virtue of the fact that the game dynamics are complicated and thus only suitable for older children, leaving younger children's responses to unfairness out of the picture.

Another consideration when implementing economic games is the type of rewards that will be used. Some work has used money with older children (e.g., participants in [23], including 7- to 10-year-olds, made decisions about points that were later converted to money). Other work with older children has used abstract rewards (e.g., tokens or points) that can be later exchanged for concrete rewards (e.g., 8- to 15-year-old participants in [32] made decision about points that were later exchanged for glow sticks). These reward types are helpful in that they closely match the rewards used in studies with adults. However, these reward types are likely more challenging for younger children who may not be as motivated by or familiar with abstract rewards. In these cases, child-friendly adaptations are required. For instance, in Wittig et al [29], 5-year-old children made decisions about gummy bears. While this change was likely necessary to maintain younger children's motivation in the task, the reward type change makes comparisons with adult data more difficult.

In summary, adapting economic games for young children while still retaining the key features to allow child-adult comparisons is inherently challenging. A key problem is that some features of the economic games must be altered to make them suitable for children. These alterations often fall into two broad categories. First, the dynamics of the games themselves need to be adapted so that children are able to understand the rules and are motivated to play. Second, and related to motivation, games with children typically use concrete rewards such as candy, toys, or stickers, while games with adults typically use abstract rewards such as money or points that can later be redeemed for a reward. Because of these two categories of changes, it can be difficult to interpret any apparently age-related differences that emerge between child and adult participants. In the next section, we discuss the Inequity Game, which we believe provides an illustrative example of the kinds of changes that are often made when adapting economic games designed for adults for use with young children.

## The Inequity Game

The Inequity Game is a task designed to test inequity aversion in children [17]. The game is loosely based on the Ultimatum Game, with the major difference being that allocations come from a third-party experimenter as opposed to one of the two players. In this task, an experimenter distributes allocations of treats (usually candy) between two peers who are sitting face-to-face on either side of an apparatus (Fig 1). The apparatus consists of two tilting trays that can be operated by a green and red handle. One of the children, the actor, uses the handles to

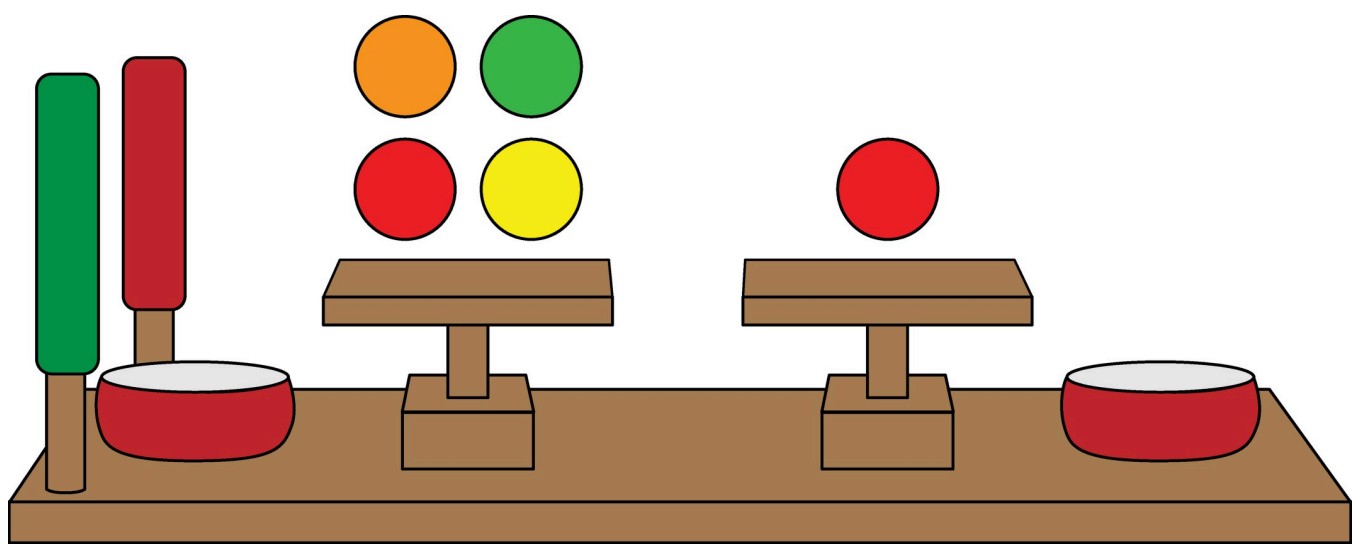

**Fig 1. Diagram of apparatus used in the Inequity Game.** Two people who do not know each other are asked to sit across from each other. The actor has access to the handles while the recipient sits passively across from the actor. The actor can accept (green handle) or reject (red handle) allocations that the experimenter places on the tilting trays. Acceptances cause rewards to be tipped into the side bowls while rejections cause rewards to be tiled into a middle bowl in which case neither participant gets them.

accept allocations by pulling the green handle or reject allocations by pulling the red handle. The recipient is passive in this game and receives payoffs based on the actor's decisions. This game assesses whether actors are more likely to reject unequal allocations than they are to reject equal allocations. Allocations can be disadvantageous from the actor's perspective—the actor receives *fewer* resources than their partner—in which case rejections provide a measure of disadvantageous inequity aversion. Alternatively, allocations can be advantageous from the actor's perspective—the actor receives *more* resources than their partner—in which case rejections provide a measure of advantageous inequity aversion. Past work with this game has shown that young children are willing to reject disadvantageous allocations [17], and this is a pattern of behavior seen across different societies [33]. However, only older children in some societies are willing to reject advantageous allocations.

While work with the Inequity Game has been informative in illustrating the developmental trajectory of inequity aversion, it is not immediately clear how comparable it is with work from adults. As stated above, the game is loosely based on the Ultimatum Game, but direct comparisons between the Inequity Game and the Ultimatum Game are not warranted because in the former children are responding to intentional unfairness from their partner while in the latter they are responding to allocations from an experimenter. From our perspective, this is a strength of the Inequity Game relative to the Ultimatum Game because it provides a cleaner test of responses to inequality per se, without potential interference from other motives such as reciprocating wrongs and without the additional demand of reasoning about the partner's intentions. Comparisons with other tasks used to measure inequity aversion in adults, such as the random allocation game used by Dawes and colleagues [12] are also problematic because their overall structure is fundamentally different. Finally, even if we were in a good position to compare between tasks directly because of similarities in structure, it is possible that these comparisons would be muddied by the fact that adult work typically uses abstract rewards such as money whereas the Inequity Game uses concrete rewards such as candy. Consequently, if we wish to make cleaner comparisons between children's and adult's responses to inequity per se, we need a test of adults' behavior in the Inequity Game. As we discuss in the

following section, such up-linkage—adapting a paradigm designed for children for use with adults—can help generate a fuller picture of humans' responses to inequity by ensuring that tasks remain as consistent as possible across participant groups.

## Drawing comparisons across different participant groups

While adult tasks are often adapted for use with children and animals (i.e., down-linkage), adaptations in the opposite direction—from animals or children to adults (i.e., up-linkage)—have recently gained traction in comparative and developmental literatures. Animal-human translation studies emphasize the adaptation of non-human animal paradigms for use with humans [34]. This bidirectional adaptation ensures that both paradigms are valid analogs of each other and results can therefore be compared [35]. For instance, two research groups [36, 37] adapted Brosnan & de Waal's [38] capuchin monkey (*Cebus apella*) inequity aversion task for use with adult participants to compare responses between monkeys and humans. Findings from these studies were mixed, with one showing evidence consistent with the capuchin response to inequity [36] and one showing evidence less consistent with the capuchin response [37]. Together, these up-linkage inequity aversion tasks have informed our understanding of the contexts in which inequity aversion—defined by the criteria outlined for animal work—is or is not revealed in adults, thereby helping us compare animal and human responses more directly.

Up-linkage from child to adult studies have been conducted across a range of domains, including Theory of Mind [39–41], spatial memory [42] and over-imitation [43]. This form of up-linkage is perhaps more straightforward than animal to human adaptations because there are more opportunities to maintain methodological equivalence (e.g., studies with both children and adults can involve verbal instructions). For instance, Lockhart and colleagues [44] used the same stimuli and broad procedure with adults and children in a study examining attitudes about boasting, and thus they were able to draw conclusions about developmental shifts in these perceptions. However, child to adult up-linkage studies, like the animal to adult studies, showcase a fundamental issue that arises when working with different participant groups: how can we ensure that participants are motivated to participate? What kinds of incentive are appropriate?

Adult decision-making studies often offer money, an abstract reward, as an incentive due to its universal value and ability to be used to acquire other rewards, while animal and child studies typically offer concrete rewards like food, toys or stickers. While perhaps trivial at first glance, these distinctions may represent a critical barrier in drawing clear comparisons across participant groups. Existing research shows important distinctions between the ways in which human adults reason about money compared to biologically-based rewards such as food (e.g., [45]). For instance, Rosati and colleagues [46] showed that human adults tested in a delay of gratification paradigm were more patient for money and less patient for food. In a later study, Rosati and Hare [47] showed a similar reward-type effect in the context of risky decision-making: adults were relatively more risk-seeking when making decisions about concrete rewards (food and prizes) and relatively less risk-seeking when making decisions about abstract rewards (money). Additionally, work with children suggests that reward type (e.g., necessary versus luxury items; [48, 49]) and children's perceived value of rewards [50] affects fairness decision making.

Taken together, the work reviewed here underscores the inherent challenges of conducting research that cuts across age (and species) boundaries. The fact that most child fairness studies use concrete rewards while most adult studies use abstract rewards may exert an important, and as yet poorly understood, influence on decision-making. Due caution must therefore be

exercised when comparing fairness behavior across child and adult tasks that vary the rewards at stake. With respect to our specific goal of understanding inequity aversion across children and adults, this work suggests that it is crucial to test developmental paradigms with adult samples in order to draw comparisons across these participant groups. To this end, the Inequity Game presents a good opportunity for examining fairness in both children and adults because (1) it provides a specific test of responses to inequity in isolation of other motives and (2) it is particularly amenable to explorations of the effects of reward type on decision-making.

## Present study

In the present study, we used the Inequity Game to test adults' responses to disadvantageous and advantageous inequity. Our broad aim in conducting this work was threefold. We aimed to (1) establish the extent to which this method can be used to tap into fairness concerns in adults as well as children; (2) explore the effects of different reward types on fairness decisions in adults; and (3) contribute to the relatively incipient literature on "up-linkage" tasks from children to adults. Our specific aims were to address the two questions outlined above: First and foremost, do adults show both forms of inequity aversion in the Inequity Game? Second, does reward type influence adults' responses to inequity? Additionally, we aimed to assess whether we see structurally similar or different responses by comparing adults' behavior in our task with past work that has used the Inequity Game with children. To these ends, we tested adults' responses to both disadvantageous and advantageous distributions of candy rewards, which are typically used when testing children in this context, *as well as* tokens that could later be traded in for money.

Past work on the Inequity Game provides hints that this kind of comparison will be revealing. In McAuliffe, Blake and Warneken [51], the Inequity Game was used to establish whether children reject inequity out of spite. As part of this study, an adult sample was tested using the candy reward typically used in the game. Adults rarely rejected inequity, and when they did, they showed a pattern opposite to what is normally observed in children: they were more likely to reject advantageous allocations of candy than disadvantageous allocations of candy. This result is puzzling and could have arisen for one of at least two reasons. The first possibility is that adults will not show disadvantageous inequity aversion in this context, one in which they must pay a small cost to deprive a partner of a large reward. If this were true, it would raise questions about what changes across development attenuate the strong response to disadvantageous inequity that we see in children in this context. The second possibility is that adults would show disadvantageous inequity aversion in this context if they were making decisions about money, a resource that has been shown to elicit disadvantageous inequity aversion in previous work (e.g., [12]). If this were true, it would inform comparisons between child and adult behavior on this task because it would suggest that reward type is a key influence on this behavior but that the inequity aversion response itself may not show major differences between children and adults. This second possibility is consistent with other work that shows, for instance, a difference between abstract and concrete rewards [46, 47] in driving decision-making in adults. These questions are important to address as they will help answer *specific* questions about this fairness task and will speak to the more *general* question about how comparable results are between adult and kid studies.

## Method

### Participants

Pairs of adults who were unfamiliar with each other were recruited in a public plaza in Cambridge, Massachusetts and each person was assigned to the role of either actor or recipient (see

below for details). Participants were 18 years or older (mean actor age = 28.11 years, standard deviation = 12.8; minimum age = 18, maximum age = 77; note we were missing age for one actor). Participants were told that participation was voluntary and that they would receive either candy or money in the game. Those who agreed were brought to a testing area, a portable table with the game apparatus. A total of 104 pairs were tested in one of two conditions: disadvantageous inequity (N = 54, 33 females) or advantageous inequity (*N* = 50, 32 females). We tested but excluded ten additional pairs (pilot participants (N = 3); out of age range (N = 1); experimenter error (N = 6)). At the time of data collection (Summer/Fall 2013) we were not collecting demographic information from participants tested in public spaces. We did, however, ask about their experience with economics and, of our 104 participants, *N* = 42 (40%) reported having taken or that they were currently taking a university-level economics course.

## Design

We employed a 2 x 2 x 2 design in which participants were assigned to one of two conditions (disadvantageous or advantageous) and, within condition, one of two reward types (Skittles or tokens). Distribution (equal or unequal) was manipulated within-subject: each pair was randomly presented with six equal trials in which the actor received one reward and the recipient received one reward (1–1) and six unequal trials. The direction of inequality depended on condition. In the disadvantageous inequity (DI) condition, pairs were presented with six trials in which the actor received one reward while the recipient received four rewards (1–4). In the advantageous inequity (AI) condition, the direction of inequality was reversed: pairs were presented with six trials in which the actor received four rewards while the recipient received one reward (4–1). Note that, due to experimenter error, one pair received only 10 trials and another pair received an extra equal trial which was deleted.

Based on previous work on the Inequity Game with children (e.g., [17, 51]), our recruitment goal was to recruit a minimum of 20 pairs per condition x reward type cell. However, our sample was ultimately imbalanced with respect to reward type such that more participants were tested with Skittles than with tokens (see Supplement for details and for additional analyses demonstrating that our results are robust when sub-setting our data to correct this imbalance).

## Procedure

The actor and recipient sat face-to-face at the game apparatus (Fig 1) and the experimenter allocated the resources on trays designated for each participant. Each testing session was randomly assigned a resource, candy or tokens. If candy was used, participants were told that they could keep any candy they acquired at the end of the game. If tokens were used, participants were told that they could trade their tokens for money at the end of the game; each token was worth 10 cents. During piloting, we asked participants if they valued the resources being used (e.g., do you like Skittles; Do you like money?). However, asking participants whether they liked money proved to be quite awkward and to cause some degree of discomfort. Because we wanted to keep our script as symmetrical as possible across reward types, we dropped this question from the procedure prior to data collection.

Participants were randomly assigned to one role using a coin: the actor, who would accept or reject the allocation of resources, and the recipient, who played a passive role. Participants played only one role (actor or recipient) in the game and participated in only one condition (DI or AI) using one resource (candy or tokens).

The experimenter explained the game to the participants and demonstrated how the apparatus worked. The decider could pull one of two handles on each trial: a green handle to accept the offer—this tilted both trays toward each participant, so the resource fell into their

respective bowls; or a red handle to reject the offer—this tilted both trays to the middle so that the resource dropped into a covered bowl and no one was able to keep the resource for that trial. The experimenter demonstrated how each handle worked with two trials, each with one resource on each tray, and stated the outcome to the participants (e.g., "you each get one [candy/token]" or "no one gets any [candy/tokens]"). The order in which the handles were demonstrated (red and green) was counterbalanced. Participants were instructed not to talk during the game.

Participants were then given three practice trials to ensure that the actor understood how each handle worked. For the practice trials, the experimenter placed different allocations of candy on the trays and prompted the actor by asking, "which handle would you like to pull?" The allocations for the practice trials were a) 1 resource each (1–1); b) 0 for the actor and 1 for the recipient (0–1; disadvantageous); and c) 1 for the actor and 0 for the recipient (1–0; advantageous). The order of the second two practice trials was random. The actor's decisions in these trials were spontaneous and not reinforced. The purpose of the practice trials was to familiarize the actors with the handles. Thus, if an actor pulled the same handle for all three trials and gained no experience with the other handle, the experimenter asked the actor to demonstrate how the handle that had not yet been pulled worked. For each practice trial and all experimental trials, the experimenter stated the outcome of the decision (e.g., "You (actor) get one resource and [recipient's name] gets none.").

Between trials, the experimenter placed a flat stick across the trays while allocating the resources. Participants were told they should wait for the experimenter to lift the stick before pulling one of the handles. Others were allowed to watch the game from a distance so as not to indirectly influence the actors' decisions, and prospective participants were told they could not watch the game so it would be a surprise.

As described above, participants in each condition received twelve trials, six equal and six unequal trials. The equal trials, consisting of one resource each for the actor and recipient (1–1), was used to control for any general tendency participants may have had to reject allocations. The unequal trials differed based on condition. In the DI condition, the unequal trials consisted of 1 resource for the actor and 4 for the recipient (1–4); in the AI condition, the unequal trials consisted of 4 resources for the actor and 1 for the recipient (4–1). The order of the twelve trials was randomized within-session. During the game, accepted rewards accumulated in bowls visible to both players. At the end of the game, and depending on the reward type used, players could either keep their candy or exchange their tokens for money.

After the testing session, the experimenter asked the actor why they made the decisions they did on the unequal trials. For instance, if they always pulled the green handle on unequal trials, they were asked why they did this; if they pulled both red and green on unequal trials, they were asked why they sometimes pulled the red handle and sometimes pulled the green handle. Participants' responses were recorded but not systematically coded and not explored further here because they are peripheral to our central research questions. When asked if they wanted to take Skittles home at the end of the task, 18 of the 62 participants tested with Skittles (29%) decided not to take the candy rewards home. We endeavored to take notes on whether participants offered to share or did share their resources after the task (recognizing, of course, that sharing may have happened after participants left the testing area). Our notes indicate that there were five such sharing events with money and twelve with Skittles, one of which was just to trade colors.

## Coding and analysis

The actors' decisions on each trial (accept or reject) were coded by the experimenter. After testing was completed, research assistants used videos to re-code all decisions for participants

who had agreed to be videotaped. Videos were available for 88% of the sessions. There was high agreement between paper and video for actors' decisions (< 2% disagreement). Disagreements were resolved by re-watching videos.

Analyses were conducted in R (Version 15.6.0; [52]). We conducted generalized linear mixed models (GLMMs) with a binary response term (reject = 1; accept = 0) using the 'lme4' package [53]. All models included participant identity as a random effect (intercepts) to control for repeated measures. In keeping with [54], we first conducted a null model which included only participant identity and compared that model to a full model which included the three-way interaction between condition (disadvantageous, advantageous), reward type (tokens, Skittles) and distribution (equal, unequal). The full model provided a better fit to our data ($\chi^2_7$ = 67.03, p < 0.001). We then tested whether results of our full model were robust to the inclusion of gender and age terms (Table 1).

We examined the predictive power of individual terms by dropping them from our model and comparing a model with the term of interest to one without it using likelihood ratio tests (LRT, using the 'drop1' command). We conducted post-hoc analyses using the 'lsmeans' package and the multivariate t (mvt) correction. Plots with binomial confidence intervals were

**Table 1. Estimates and bootstrapped CIs (N = 1,000 simulations) of fixed effects in Generalized Linear Mixed Models predicting children's rejections (= 1) in the Inequity Game.** Baselines were set as follow: Condition: AI; Distribution = Equal; Resource type = Skittles; Actor gender = Female. Table also shows goodness-of-fit statistics.

| | Full model | Full model with gender and age | Disadvantageous | Advantageous |
|---|---|---|---|---|
| (Intercept) | -3.70* | -4.90* | -3.40* | -7.12* |
| | [-5.28; -2.67] | [-6.75; -3.25] | [-5.41; -1.54] | [-11.77; -4.73] |
| Condition: DI | 0.93 | 0.46 | | |
| | [-0.57; 2.65] | [-1.25; 1.97] | | |
| Distribution: Unequal | 1.93* | 1.95* | 0.15 | 2.04* |
| | [1.20; 2.90] | [1.18; 2.86] | [-0.55; 0.87] | [1.28; 3.13] |
| Resource type: Tokens | -1.53 | -1.54 | -1.51* | -1.65 |
| | [-8.32; 0.44] | [-6.78; 0.33] | [-3.76; -0.14] | [-8.12; 0.51] |
| Condition x Distribution | -1.73* | -1.79* | | |
| | [-2.92; -0.70] | [-2.81; -0.68] | | |
| Condition x Resource type | -0.35 | 0.23 | | |
| | [-3.74; 6.35] | [-2.43; 5.05] | | |
| Distribution x Resource type | 0.03 | 0.01 | 1.91* | -0.02 |
| | [-1.53; 6.34] | [-1.25; 3.95] | [0.67; 3.45] | [-1.72; 5.61] |
| Condition x Distribution x Resource type | 1.91 | 1.97 | | |
| | [-4.24; 4.52] | [-2.43; 4.30] | | |
| Actor age | | 0.05* | 0.02 | 0.13* |
| | | [0.01; 0.10] | [-0.02; 0.07] | [0.05; 0.25] |
| Actor gender: Male | | -0.18 | 0.19 | -0.49 |
| | | [-1.59; 1.02] | [-1.13; 1.40] | [-2.55; 1.60] |
| AIC | 842.08 | 831.61 | 469.66 | 361.43 |
| BIC | 888.22 | 887.91 | 500.85 | 392.18 |
| Log Likelihood | -412.04 | -404.80 | -227.83 | -173.71 |
| Number of trials | 1246 | 1234 | 636 | 598 |
| Number of participants | 104 | 103 | 53 | 50 |
| Variance: ID (Intercept) | 5.16 | 4.72 | 3.29 | 5.94 |

* 0 outside the confidence interval

initially produced in R using ggplot [55] and the Agresti-Coull method [56] and were then reproduced in Microsoft Excel. In the final step of our analyses, we conducted a qualitative, visual comparison of data from the current study to data from a previous study that used the same task [51].

In addition to our main analyses of interest, we conducted a set of exploratory analyses examining whether actors' decisions varied across trials. We tested this both by exploring whether decisions varied across all 12 trials and whether decisions varied across trials of each distribution type (i.e., did decisions vary across the 6 presentations of unequal trials and/or the 6 presentations of equal trials). These additional analyses are included in our SOM.

## Ethical note

This study was approved by the Harvard University IRB (F18470). Participants provided written informed consent prior to participation. No deception was used.

## Results

Our first question was whether adult participants would show inequity aversion in the Inequity Game, a task originally designed for children. As Fig 2 shows, adults did preferentially reject unequal allocations, therefore showing some inequity aversion in this task. However, these rejections were moderated by both condition and reward type. Our GLMM including the

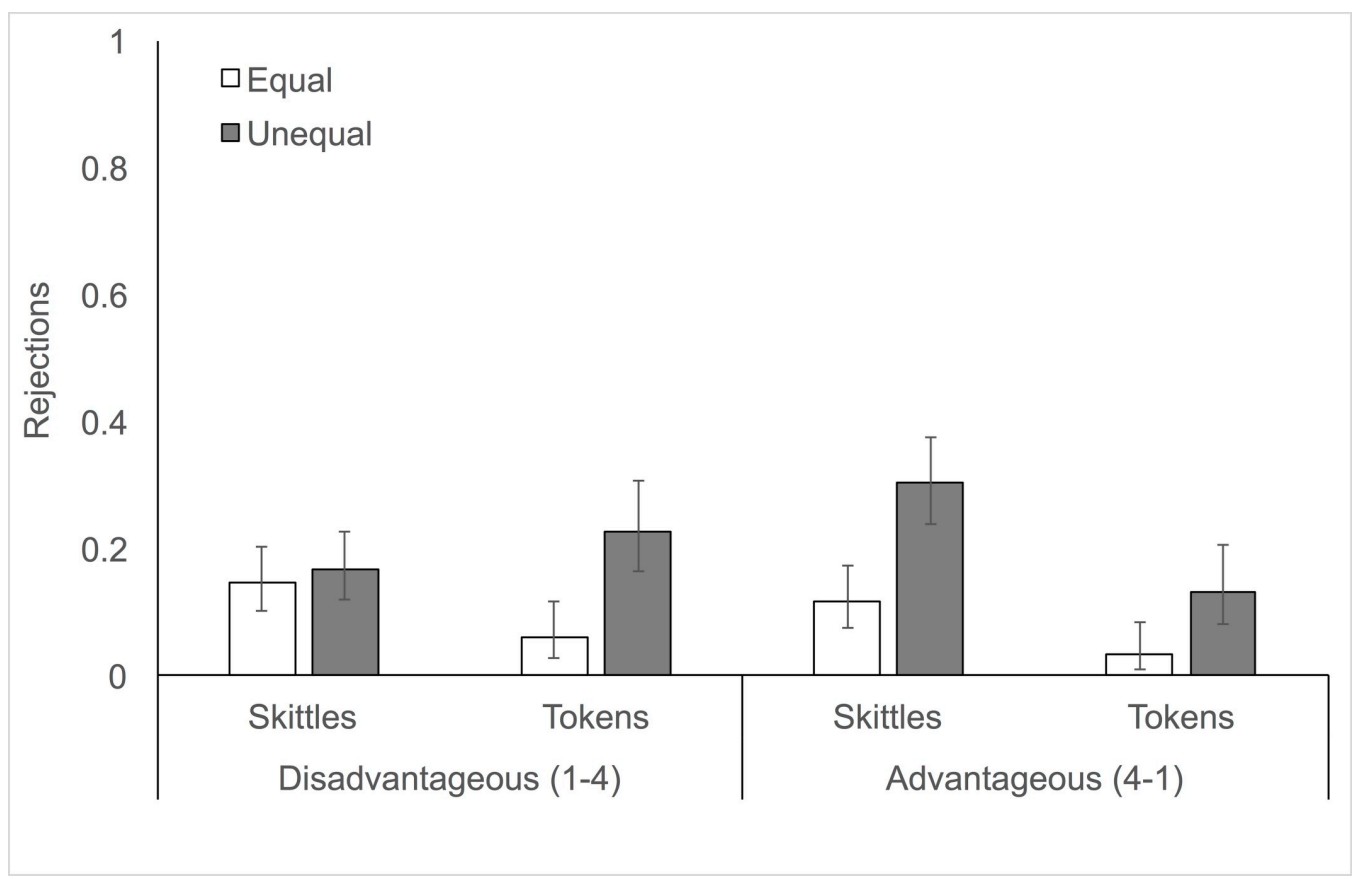

**Fig 2. Proportion of rejections of equal and unequal allocations in disadvantageous and advantageous conditions.** Rejections are shown by reward type. Error bars show binomial confidence intervals.

three-way interaction between condition, reward type and distribution provided a marginally better fit to the data than a model without this term (LRT, $\chi^2_1 = 3.68$, $p = 0.055$). As Fig 2 suggests, this interaction is due to the fact that participants showed relatively stronger inequity aversion when deciding about tokens in the disadvantageous condition and when deciding about candy in the advantageous condition. Note that this finding is robust to the inclusion of gender and age (LRT, $\chi^2_1 = 3.88$, $p = 0.049$). Models with age and gender included a sample of $N = 103$ as opposed to our full sample of $N = 104$ because we were missing age for one participant. Our model including age also found that age was a predictor of rejections (LRT, $\chi^2_1 = 5.28$, $p = 0.022$): participants were increasingly likely to reject with age (Table 1; S2 Fig in S1 File). To further unpack the relationship between reward type and inequity aversion, we ran subsequent analyses divided by condition as is customary when examining data from the Inequity Game [17, 33, 51].

## Effects of reward type on disadvantageous inequity aversion

When we examined how reward type affected participants' rejections of disadvantageously unequal trials relative to equal trials, we found that our GLMM including the interaction between reward type and distribution provided a better fit to our data than one without this term (LRT, $\chi^2_1 = 11.49$, $p < 0.001$). Note that this model also included gender and age terms, which were not significant predictors ($ps > 0.3$). Post-hoc comparisons (with mvt correction) of rejections of unequal versus equal trials within reward type revealed that participants showed disadvantageous inequity aversion—i.e., they rejected more disadvantageous than equal allocations—when making decisions about tokens ($b = -2.06$, $p = 0.0002$) but not when making decisions about Skittles ($b = -0.2$, $p = 0.91$). These differences are illustrated clearly in Fig 2.

## Effects of reward type on advantageous inequity aversion

When we examined how reward type affected participants' rejections of advantageously unequal trials relative to equal trials, we found that our GLMM including the interaction between reward type and distribution did not provide a better fit to our data than one without this term (LRT, $\chi^2_1 \sim 0$, $p = 1$). As above, this model also included gender and age terms and age was a significant predictor of rejections (LRT, $\chi^2_1 = 7.49$, $p = 0.006$). As Fig 2 illustrates, participants showed advantageous inequity aversion—i.e., they rejected more unequal than equal allocations—when making decisions about both tokens and Skittles (post-hoc comparisons with mvt correction: Skittles, $b = -2.03$, $p < 0.001$; tokens, $b = -2.03$, $p = 0.012$).

## Qualitative comparisons with past work with adults and children

We next turn to the question of how participants in this task compare to child and adult participants who have been tested in previous versions of this task. We compared data from the current task to the only previously-run version of the task in which adult participants were tested [51]. Specifically, we used data from the condition of McAuliffe et al. [51] in which rejections affect payoffs for both actor and recipient, corresponding to the condition of our current study.

Fig 3 suggests that when adults are presented with Skittles, they are relatively more likely to show advantageous inequity aversion than disadvantageous inequity aversion. Moreover, adults show overall lower levels of disadvantageous inequity aversion and advantageous inequity aversion than children for both types of rewards.

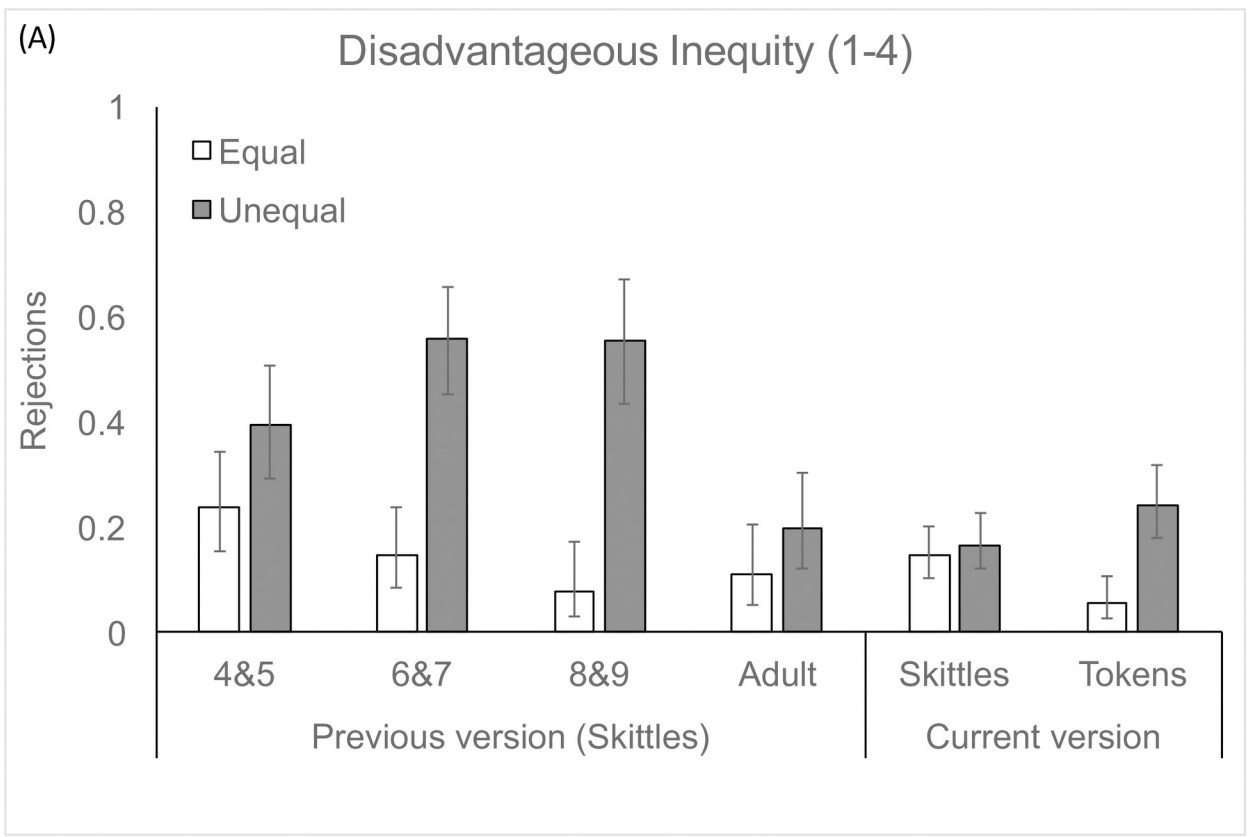

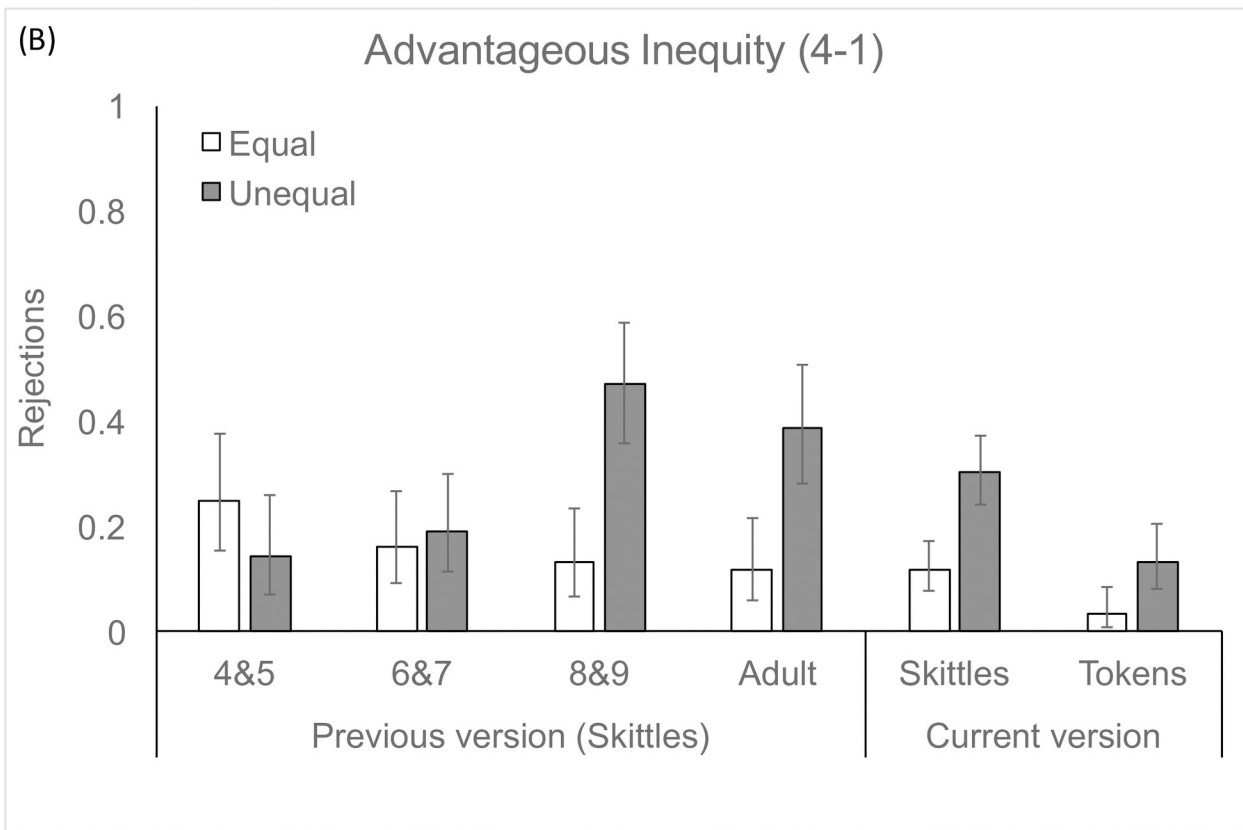

**Fig 3. Comparisons between the data reported here ("Current version") and data from McAuliffe et al., ([51]; "Previous version").** Data are shown by Disadvantageous (A) and Advantageous (B) conditions. Comparisons indicate that we replicate the pattern seen previously in which adults preferentially reject advantageous, but not disadvantageous, allocations when presented with concrete rewards (Skittles). Our new data add to this picture showing that, when presented with abstract rewards (tokens that can be traded in for money), adults show both forms of inequity aversion.

## Discussion

Our study provides a test of inequity aversion in adults using a task originally designed for young children. Using this task, we additionally explored whether inequity aversion in both disadvantageous and advantageous directions is influenced by reward type. Our findings show that, like older children in previous studies, adults reject both disadvantageous and advantageous inequity. However, their rejections are importantly moderated by reward type. Specifically, adults showed stronger disadvantageous inequity aversion when making decisions about abstract rewards (money) than about concrete rewards (candy). By contrast, they showed advantageous inequity aversion when making decisions about both abstract and concrete rewards. When we compared adults' responses to children's responses on the same task from past work, a notable difference is that children's responses to disadvantageous inequity are relatively stronger than those of adults, suggesting that a response to disadvantageous inequity may become attenuated over development.

First and foremost, we view this study as an important contribution to the literature because it provides—at the very least—an indirect comparison between children's and adults' responses to unfairness in the exact same paradigm. Inequity aversion has been difficult to compare directly in children and adults due to fundamental differences in the methods used to test these different age groups. For instance, comparisons between children's behavior in the Inequity Game and adults' behavior in the Ultimatum Game are unwarranted because (1) the Ultimatum Game involves but is not limited to responses to inequality and (2) adult work with the Ultimatum Game typically involves abstract rather than concrete rewards. It is encouraging to learn that adults, like children, respond to inequity in this task, as it then allows us to consider the fuller developmental trajectory of this response and presents a method which can now be used to test inequity aversion in participants across ages.

One interesting aspect of our findings is that we see a clear asymmetry between disadvantageous and advantageous inequity aversion in adults and children. Specifically, here, we found that reward type influences adults' responses to inequity, with disadvantageous inequity aversion emerging more strongly in the context of abstract rewards and advantageous inequity aversion emerging equally across the contexts of abstract and concrete rewards. The asymmetry between disadvantageous and advantageous inequity aversion that is apparently preserved across wide age-ranges is further evidence that these types of inequity aversion are distinct and likely the result of different psychological processes, adding to the findings showing differences in the age of emergence, cross-cultural variation, and neural underpinnings (reviewed in [10]). An interesting question that emerges from these findings is *why* we see this reward type effect in the context of disadvantageous but not advantageous inequity.

One potential explanation for the observed effect of reward type on disadvantageous inequity aversion is that adults valued money more than candy. This could explain why they did not show disadvantageous inequity aversion when presented with candy, but did when presented with tokens: seeing a partner with more candy was relatively less aversive than seeing a partner with more tokens. However, while relative reward value could offer a partial explanation, we do not believe it can fully account for our findings. If reward value drives decision-making in the Inequity Game, it stands to reason that we would have seen an interaction of reward type and distribution in both the disadvantageous and the advantageous conditions.

However, although the absolute rate of rejections was higher in the Advantageous condition when participants were presented with candy than with tokens (Fig 2), we did not see an interaction between reward type and distribution as we did in the Disadvantageous condition. Why, then, would reward type have influenced disadvantageous but not advantageous inequity aversion? One possibility is that adults viewed rejections of disadvantageous candy distributions as a relatively cheap way to do something nice for their partner—i.e., to deliver a relatively larger reward (four resources) to their partner at a relatively low cost (one resource) to themselves. Of course, an alternative possibility is that the interaction effect between reward type and distribution is weaker in the context of advantageous inequity than in the context of disadvantageous inequity, and we may have been underpowered to detect the effect (see SOM for sensitivity analyses). In discussing how the different rewards were valued by adult participants, it is worth noting that the stake sizes used in the study were generally quite low, potentially decreasing the importance of individual decisions in this task. Indeed, low valuation of the resources used in this task may reflect an important difference in how adults versus children approach decision-making in this context. For children, the rewards used may generally be valued more and thus each decision may feel more meaningful. While we view this as a possibility, it is also worth noting that stake size does not tend to be a major determinant of people's behavior in economic games [57, 58]. In the next sections, we turn to a more detailed discussion of how these results compare to similar work with children.

We now turn to the question of how adults' responses in the Inequity Game compare with children's responses in the exact same task. Our qualitative comparison with children's inequity aversion (Fig 3) revealed an interesting pattern of similarities and differences. When looking at the disadvantageous inequity condition, we see that relative to children, adults were less likely to reject disadvantageous inequity, even when making decisions about money. There are at least two reasons why this may be the case, both of which warrant further exploration. First, it may demonstrate a difference in how adults weigh efficiency and fairness, preferring to see rewards distributed to both players rather than being wasted in the name of fairness. In the Ultimatum game, many adults favor fairness over efficiency and readily reject when proposers make unfair offers. By contrast, because unequal allocations in the Inequity Game are generated by a third party (the experimenter) as opposed to coming from another player, questions of motives and intentionality are off the table. Therefore, perhaps adults care more about disadvantageous allocations when they arise via selfish intent, as is the case in the Ultimatum Game, than the distribution alone, as in the Inequity Game. We view this as an intriguing possibility which could be tested by conducting a direct comparison between adults' rejections in the Inequity Game compared to the Ultimatum Game. Given that previous work has shown that it is not until later in development that children and adolescents begin to integrate information about intentionality into their Ultimatum Game rejections [23, 32], we would predict that while younger children would show similar levels of disadvantageous inequity across Inequity Game and Ultimatum Game, older children and adolescents would distinguish between these contexts by showing fewer rejections in the Inequity Game than the Ultimatum Game (for a discussion of intentionality effects on responses to unfairness in the context of punishment see [59]). A second possible reason for the difference between adults' and children's responses to disadvantageous inequity is that, as we suggest above, adults may have viewed accepting disadvantageous allocations as a relatively cheap way to do something nice for their partners. Younger children, on the other hand, tend to be spiteful in this context [51], preferentially rejecting disadvantageous allocations when doing so deprives their peer of a reward. An interesting and open question is whether these spiteful responses decrease over developmental time, giving way to a more prosocial response in this relatively low-cost context.

When we compare adults and children in the advantageous condition, we again see an attenuation of response across ages, but one that is not as extreme as in the disadvantageous condition. These data thus suggest that adults, like older children, are willing to pay a relatively large cost to prevent the delivery of a distribution that disadvantages a partner. Data from sessions in which adults made decisions about candy show nice convergence with previous work, replicating the finding that adults are willing to reject advantageous allocations of Skittles [51]. Here we add to this finding by showing that they are additionally willing to reject advantageous allocations of money. If we assume that adults care more about money than Skittles, this provides evidence that rejections of advantageous allocations are not the result of lack of motivation for the rewards at stake. Rather, other factors must be contributing to adults' willingness to reject in this context. One possibility is that adults do not want to *appear* unfair to the experimenter and/or their unfamiliar partner. Consistent with this possibility, previous work has found that advantageous inequity in older children is stronger when an experimenter [60] or peer partner [61] knows about the decider's advantage.

From a methodological perspective, our study showcases the utility of conducting up-linkage tasks as a tool for understanding how children and adults respond when presented with the exact same paradigm. We show that the Inequity Game provides a means of testing inequity aversion in adults and we suggest that this could be used as an alternative to the Ultimatum Game in cases in which people are interested in studying inequity in isolation from other motives. Our data also suggest that it is important to consider the *kinds* of resources used in economic games. Here we found a difference between adults' responses to abstract and concrete rewards in the disadvantageous condition. This was important because had we not included an abstract reward manipulation in addition to the concrete reward manipulation, we might have erroneously concluded that the Inequity Game does not elicit disadvantageous inequity aversion in adults.

There are several limitations of the current work, a number of which point to clear avenues for future work. First, our sample size was on the small side and thus constrained the analyses we were well-powered to conduct. It would be useful to conduct future lines of this work with a larger sample so that interaction between condition, resource type and other variables such as participant age and gender can be examined. Although these were not the focus of our experimental design, we view them as interesting questions that are worthy of exploration. Participant gender, in particular, would be interesting to explore in future studies. For instance, does inequity aversion depend on the gender composition of participant pairs? Specifically, are people more likely to reject unfair allocations when paired with a same- vs other-gendered peer? Future work could systematically vary the gender composition of pairs to examine this and related questions. Additionally, with a bigger sample, individual-level predictors (e.g., personality measures) could be included in analyses to examine whether they help explain variation in inequity aversion across condition and resource type. Second, we were not in a position to explore participants' explicit justifications for their decisions because it is not clear that responses were always recorded verbatim by our live coders. Additionally, responses could not be checked from many of the videos due to surrounding noise. In future work, it would be interesting to record and analyze participants' justifications, potentially capitalizing on the existence of natural language tools to facilitate this kind of analysis. Third and finally, rejections were relatively infrequent in this study. These low rates of rejection are potentially reflective of actual levels of inequity aversion in our sample. Alternatively, low rejection rates could be due to a number of features of our design, which include but are not limited to the following: (1) we recruited participants in a public space and tested those who were willing to participate. In doing so, we may have biased our sample in favor of a certain type of participant. For instance, perhaps our sampling resulting in the recruitment of participants who were

particularly averse to depriving their partner of resources and thus were unlikely to reject; (2) participants were tested face-to-face and there may thus have been nonverbal influences on behavior, and because of this face-to-face design, features of the recipient may have been particularly salient to actors; (3) the experimenter was responsible for allocating distributions which introduces a third party to the interaction. While the role of the experimenter has been explored in work with children [62] and these results suggest that rejections are not directed toward the experimenter, the influence of the experimenter cannot be ruled out in the present study.

What steps should be taken next to help us gain a clearer picture of how reward type influences inequity aversion? In addition to tackling the limitations listed above, future research should seek to explain *why* reward type influences inequity aversion, at least in the context of disadvantageous inequity aversion. To this end, future work in this area could focus on assessing individual-level valuation of different rewards to explore whether perceived reward value drives these effects. Additionally, studies testing these effects would benefit from assessing valuation of rewards in both adults and children within a single study to be able to more directly compare inequity aversion and the influence of reward types across age groups.

In sum, our findings demonstrate that adults show both disadvantageous and advantageous inequity aversion in the Inequity Game, a task designed for children. This study thus validates the use of this paradigm for work with a broader age range and allows us to compare children's and adults' responses to inequity. Consistent with past work, we find an asymmetry in responses to inequity in different directions. Here, this asymmetry is dependent on reward type, with adults showing disadvantageous inequity aversion when presented with abstract rewards but not concrete rewards and showing advantageous inequity aversion when presented with both abstract and concrete rewards. Our findings highlight inequity aversion as a response seen across ages: even in cases in which unequal allocations are generated by a third-party, people across ages reject them at personal cost, showcasing the strength of humans' aversion to unequal outcomes.

## Supporting information

**S1 File. Supplementary Online Material (SOM) file.**
(PDF)

**S1 Data. Raw study data.**
(CSV)

**S2 Data. Random subset.**
(CSV)

**S3 Data. Raw data from child study.**
(CSV)

**S4 Data. Analysis code (R).**
(R)

**S5 Data. Simr data for sensitivity analyses.**
(CSV)

## Acknowledgments

We are grateful first and foremost to the people who agreed to participate in this study. Additionally, we are grateful to those who helped collect, code and process data for this study.

Specifically, we would like to thank Matthias Beringer, Hannah Bolotin, Gorana Gonzalez and Shaun O'Grady.

## Author Contributions

**Conceptualization:** Katherine McAuliffe, Natalie Benjamin, Felix Warneken.

**Data curation:** Katherine McAuliffe, Natalie Benjamin.

**Formal analysis:** Katherine McAuliffe.

**Investigation:** Katherine McAuliffe, Natalie Benjamin.

**Methodology:** Katherine McAuliffe, Natalie Benjamin, Felix Warneken.

**Project administration:** Katherine McAuliffe, Felix Warneken.

**Resources:** Felix Warneken.

**Supervision:** Felix Warneken.

**Writing – original draft:** Katherine McAuliffe, Natalie Benjamin.

**Writing – review & editing:** Katherine McAuliffe, Natalie Benjamin, Felix Warneken.

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
