## [Decision Letter · Decision Letter 0]

17 Aug 2021

PONE-D-21-21504

Reward type influences adults’ rejections of inequality in a task designed for children

PLOS ONE

Dear Dr. McAuliffe,

Thank you for submitting your manuscript to PLOS ONE. After careful consideration, we feel that it has merit but does not fully meet PLOS ONE’s publication criteria as it currently stands. Therefore, we invite you to submit a revised version of the manuscript that addresses the points raised during the review process.

You will find two reports (one is attached as a separated file). Both reviewers provide a number of comments and suggestion to clarify certain concept and to improve the paper. Personally I liked the reports very much. They are extremely professional and provide a lot of suggestions -ene of the referees even provided STATA code! I would say that the general tone is fairly positive.

We look forward to receiving your revised manuscript.

Kind regards,

Pablo Brañas-Garza, PhD Economics

Academic Editor

PLOS ONE

Journal Requirements:

2. Please provide additional details regarding participant consent. In the Methods section, please ensure that you have specified (1) whether consent was informed and (2) what type you obtained (for instance, written or verbal). If your study included minors, state whether you obtained consent from parents or guardians. If the need for consent was waived by the ethics committee, please include this information.

Reviewers' comments:

Reviewer's Responses to Questions

**Comments to the Author**

1. Is the manuscript technically sound, and do the data support the conclusions?

Reviewer #1: Yes

Reviewer #2: Partly

2. Has the statistical analysis been performed appropriately and rigorously? 

Reviewer #1: No

Reviewer #2: I Don't Know

3. Have the authors made all data underlying the findings in their manuscript fully available?

Reviewer #1: Yes

Reviewer #2: Yes

4. Is the manuscript presented in an intelligible fashion and written in standard English?

Reviewer #1: Yes

Reviewer #2: Yes

5. Review Comments to the Author

Reviewer #1: I think that the results section should be revised. See point 5 in the attached file for further details. Also, the paper will improve if in the discussion section authors dicuss some of the caveats I include on the previous points.

Reviewer #2: Reward type influences adults’ rejections of inequality in a task designed for children

The paper shows an interesting experiment 2x2x2 about inequality, a mini dictator game (iterated), with 104 decision makers conducted in 2013, for its realization they use a contraption, the machine is very ingenious, and we think it adds a plus to the experiment, in which players can participate regardless of their age, despite that the age criteria of this experiment are subjects over 18 years old.

The actors make two decisions, (1,1 vs 0,0) and (1,4 or 4,1 vs 0,0) and each decision is made 6 times, all decision was paid, and the individual has direct information about the accumulated amount

We would need to read information about statistical power analysis.

In the regression model we would like to see different models in which you add the iterations one by one and increase the number of controls by adding the information you have available about the recipient. We would like to look at the p-values in the regression table and you must add cluster for individuals. You must add the analysis code to see the replication when performing an analysis with random values.

They don’t analyze the consistency in the decisions. Including inconsistent subjects in the analysis implies that we do not have a model that includes rounds decisions. And maybe the study could show if the accumulated amount modifies the result. We would add the analysis of the individual's first decision only, to show the robustness of the model.

We find it interesting if you have the explanations of the individuals when making the changes in their decisions, to analyze that motivation, using a Natural Language tool.

Regarding the comparison of the children's data, the data are not available in the additional information you provide so we do not have enough information to review that analysis. we would like to see a more aggregate behavior of the children's data compared to an aggregate of the adult data.

6. PLOS authors have the option to publish the peer review history of their article (what does this mean?). If published, this will include your full peer review and any attached files.

Reviewer #1: No

Reviewer #2: No

---

## [Decision Letter · Decision Letter 1]

17 Feb 2022

PONE-D-21-21504R1Reward type influences adults’ rejections of inequality in a task designed for childrenPLOS ONE

Dear Dr. McAuliffe,

Thank you for submitting your manuscript to PLOS ONE. After careful consideration, we feel that it has merit but does not fully meet PLOS ONE’s publication criteria as it currently stands. Therefore, we invite you to submit a revised version of the manuscript that addresses the points raised during the review process. Reviewer #1 has already accepted the paper while  #2 is asking for 2 minimal amends. Please add the requested information and I will accept the paper without any additional round. Thanks, Pablo

We look forward to receiving your revised manuscript.

Kind regards,

Pablo Brañas-Garza, PhD

Academic Editor

PLOS ONE

Journal Requirements:

Reviewers' comments:

Reviewer's Responses to Questions

**Comments to the Author**

1. If the authors have adequately addressed your comments raised in a previous round of review and you feel that this manuscript is now acceptable for publication, you may indicate that here to bypass the “Comments to the Author” section, enter your conflict of interest statement in the “Confidential to Editor” section, and submit your "Accept" recommendation.

Reviewer #1: All comments have been addressed

Reviewer #2: All comments have been addressed

2. Is the manuscript technically sound, and do the data support the conclusions?

Reviewer #1: Yes

Reviewer #2: Partly

3. Has the statistical analysis been performed appropriately and rigorously? 

Reviewer #1: Yes

Reviewer #2: Yes

4. Have the authors made all data underlying the findings in their manuscript fully available?

Reviewer #1: Yes

Reviewer #2: Yes

5. Is the manuscript presented in an intelligible fashion and written in standard English?

Reviewer #1: Yes

Reviewer #2: Yes

6. Review Comments to the Author

Reviewer #1: I think authors have addressed properly all questions. Two minor caveats (for further work):

* Sorry for the confusion with blocks/allocations. However, as this implies that there is a trial order effect (that authors now consider in the Sup. Mat.), I suggest that for further research authors use the block design (E + Unequal), randomizing block order instead of all decisions. I think this could help in the reduction of contamination between trials.

* However, defining correctly (I hope) the equal/unequal decisions, and running a simple regression, design variables can explain differences in rejection rates (Unequal- Equal). Gender variables now become irrelevant:

gen t1decE= t1dec if t1dist=="e"

gen t2decE= t2dec if t2dist=="e"

gen t3decE= t3dec if t3dist=="e"

gen t4decE= t4dec if t4dist=="e"

gen t5decE= t5dec if t5dist=="e"

gen t6decE= t6dec if t6dist=="e"

gen t7decE= t7dec if t7dist=="e"

gen t8decE= t8dec if t8dist=="e"

gen t9decE= t9dec if t9dist=="e"

gen t10decE= t10dec if t10dist=="e"

gen t11decE= t11dec if t11dist=="e"

gen t12decE= t12dec if t12dist=="e"

recode t1decE t2decE t3decE t4decE t5decE t6decE t7decE t8decE t9decE t10decE t11decE t12decE (missing = 0)

gen equal = t1decE+ t2decE+ t3decE+t4decE+t5decE+t6decE+t7decE+t8decE+t9decE+ t10decE+ t11decE+ t12decE

gen unequal = t1dec+ t2dec+ t3dec+t4dec+t5dec+t6dec+t7dec+t8dec+t9dec+ t10dec+ t11dec+ t12dec_paper – equal

regress dif actorage res own_ad resXowm female other_female same_gender econ other_econ

Source | SS df MS Number of obs = 102

-------------+---------------------------------- F(9, 92) = 1.26

Model | 1.01653388 9 .112948209 Prob > F = 0.2705

Residual | 8.26015451 92 .089784288 R-squared = 0.1096

-------------+---------------------------------- Adj R-squared = 0.0225

Total | 9.27668839 101 .0918484 Root MSE = .29964

dif | Coef. Std. Err. t P>|t| [95% Conf. Interval]

-------------+----------------------------------------------------------------

actorage | .0010817 .0026372 0.41 0.683 -.004156 .0063194

res | .1847137 .0912823 2.02 0.046 .0034191 .3660083

own_ad | .1844242 .0860322 2.14 0.035 .0135569 .3552916

resXowm | -.2818712 .1282748 -2.20 0.031 -.536636 -.0271065

female | .0879401 .0679329 1.29 0.199 -.0469806 .2228608

other_female | .0425448 .0670059 0.63 0.527 -.0905347 .1756242

same_gender | .0374303 .066879 0.56 0.577 -.0953972 .1702577

econ | -.0407902 .0607982 -0.67 0.504 -.1615407 .0799603

other_econ | .0724631 .0645547 1.12 0.265 -.0557481 .2006743

_cons | -.1374377 .1312655 -1.05 0.298 -.3981422 .1232668

(still Stata, but I hope the code is easy to follow)

Reviewer #2: Thank you very much for considering the suggestions provided, and thank you very much for providing the data, which makes the review work much more enjoyable.

Just a few additional suggestions.

- In table 1.

Add in column 2, a control (endowment) for the number of tokens accumulated at the time of the decision, because that information is held by the actor in his bowl when making the decision and therefore may affect the decision.

Please add the control age and gender, in columns 3 and 4, because in column 2 we show that those controls are important.

- Also, as a foodnote, I think you could report the coefficients of the Romano-Wolf multiple hypothesis correction test, because when multiple hypothesis tests are considered simultaneously, standard statistical techniques will lead to over-rejection of the null hypotheses.

7. PLOS authors have the option to publish the peer review history of their article (what does this mean?). If published, this will include your full peer review and any attached files.

Reviewer #1: No

Reviewer #2: No

---

## [Decision Letter · Decision Letter 2]

7 Jul 2022

PONE-D-21-21504R2Reward type influences adults’ rejections of inequality in a task designed for childrenPLOS ONE

Dear Dr. McAuliffe,

Thank you for submitting your manuscript to PLOS ONE. After careful consideration, we feel that it has merit but does not fully meet PLOS ONE’s publication criteria as it currently stands. Therefore, we invite you to submit a revised version of the manuscript that addresses the points raised during the review process.

While the reviewers are now satisfied your manuscript it ready for publication our internal checks have raised a query. We notice your IRB approval number (F18470) has been used in at least two other published papers:

McAuliffe Katherine, Blake Peter R. and Warneken Felix 2014 Children reject inequity out of spiteBiol. Lett.102014074320140743

J.J. Jordan, K. McAuliffe, F. Warneken Development of in-group favoritism in children’s third-party punishment of selfishness Proceedings of the National Academy of Sciences, 111 (35) (2014), pp. 12710-12715 JSTOR

Please can you confirm all these studies are covered by the same ethics approval. Please also upload a copy of your ethical approval document.

We look forward to receiving your revised manuscript.

Kind regards,

Thomas Phillips, PhD

Staff Editor

PLOS ONE

Journal Requirements:

Reviewers' comments:

Reviewer's Responses to Questions

**Comments to the Author**

1. If the authors have adequately addressed your comments raised in a previous round of review and you feel that this manuscript is now acceptable for publication, you may indicate that here to bypass the “Comments to the Author” section, enter your conflict of interest statement in the “Confidential to Editor” section, and submit your "Accept" recommendation.

Reviewer #1: All comments have been addressed

Reviewer #2: All comments have been addressed

2. Is the manuscript technically sound, and do the data support the conclusions?

Reviewer #1: Yes

Reviewer #2: Yes

3. Has the statistical analysis been performed appropriately and rigorously? 

Reviewer #1: Yes

Reviewer #2: Yes

4. Have the authors made all data underlying the findings in their manuscript fully available?

Reviewer #1: Yes

Reviewer #2: Yes

5. Is the manuscript presented in an intelligible fashion and written in standard English?

Reviewer #1: Yes

Reviewer #2: Yes

6. Review Comments to the Author

Reviewer #1: As in the previous version I already considered the paper as aceptable in Plos One, of course, I find it acceptable now.

Reviewer #2: The authors have responded satisfactorily to all comments and the manuscript, Reward type influences adults' rejections of inequality in a task designed for children, is now ready for publication.

7. PLOS authors have the option to publish the peer review history of their article (what does this mean?). If published, this will include your full peer review and any attached files.

Reviewer #1: No

Reviewer #2: No

---

## [Editor Report · Decision Letter 3]

26 Jul 2022

Reward type influences adults’ rejections of inequality in a task designed for children

PONE-D-21-21504R3

Dear Dr. McAuliffe,

We’re pleased to inform you that your manuscript has been judged scientifically suitable for publication and will be formally accepted for publication once it meets all outstanding technical requirements.

Kind regards,

James Mockridge

Staff Editor

PLOS ONE

---

## [Editor Report · Acceptance letter]

8 Aug 2022

PONE-D-21-21504R3 

Reward type influences adults’ rejections of inequality in a task designed for children 

Dear Dr. McAuliffe:

I'm pleased to inform you that your manuscript has been deemed suitable for publication in PLOS ONE. Congratulations! Your manuscript is now with our production department. 

Kind regards, 

on behalf of

Dr James Mockridge 

Staff Editor

PLOS ONE